# Assessment of the Reduction in Vehicles Emissions by Implementing Inspection and Maintenance Programs

**DOI:** 10.3390/ijerph17134730

**Published:** 2020-07-01

**Authors:** José I. Huertas, Antonio E. Mogro, Alberto Mendoza, María E. Huertas, Rolando Ibarra

**Affiliations:** 1Tecnologico de Monterrey, Escuela de Ingeniería y Ciencias, Ave. Eugenio Garza Sada 2501, Monterrey 64849, NL, Mexico; a01366355@itesm.mx (A.E.M.); mendoza.alberto@tec.mx (A.M.); maria.huertas@tec.mx (M.E.H.); 2Camara de la Industria de Transformacion de Nuevo Leon, Parque Fundidora 501 L-95A, Monterrey 64010, NL, Mexico; ribarrar@outlook.com

**Keywords:** I/M program, RSD, Road transportation, Real-world vehicle emissions, Mexico

## Abstract

To improve air quality in urban centers, vehicle Inspection and Maintenance (I/M) programs were created to identify highly polluting vehicles and force them to undergo mechanical maintenance. In this context, a methodology, based on a single measurement campaign using a Remote Sensing Device (RSD), is presented in this work to assess the reduction in vehicles emissions obtained from implementing I/M programs. As a case study, an RSD campaign was carried out in Mexico, specifically in Monterrey’s Metropolitan Area (MMA). Approximately 0.4% of the vehicles registered in this region were sampled under similar conditions to those found in I/M programs. The results obtained suggested that 39% of the vehicles would not comply with the current national regulations for circulating vehicles. Following a conservative scenario, the implementation of a vehicle I/M program in this urban center has the potential of reducing the current mass emissions of HC, CO and NO in approximately 69%, 42% and 28%, respectively.

## 1. Introduction

Air quality exceeds the World Health Organization (WHO) recommended safe thresholds in most urban centers for human exposure [1], with the road transport sector as the primary pollutant emitter [2]. Moreover, recent studies reported that the vehicles tailpipe CO_2_ emitted under real-world conditions are up to 60 percent higher than those reported by the automotive manufacturing companies in the type-approval protocols [3]. The mechanical condition of the vehicles is considered a highly influential factor within this discrepancy. Inspection and Maintenance (I/M) programs were created with the objective of improving air quality by periodically detecting highly polluting vehicles that come from inadequate mechanical maintenance [4]. The I/M test procedure, imposition and frequency are regionally dependent as they consider the ease of implementation and the local needs.

Several studies have evaluated the effectiveness of the I/M programs in reducing the vehicle emissions. The majority of these studies are based on data directly obtained from implemented I/M programs within the United States of America. Wenzel et al. (1999) reported that the enhanced I/M program implemented in Arizona (based on the driving cycle IM240) reduced the HC, CO and NO_X_ emissions by 16%, 17% and 7%, respectively [5]. DeHart et al. (2002) followed a different approach using on-road data obtained with several measurement campaigns using Remote Sensing Devices (RSD) from I/M and non-I/M fleets of vehicles taken within the first two years of the Atlanta’s enhanced I/M program. With the assumption that the on-road emissions differences are related only to the I/M program, they estimated a reduction in the CO emissions by 26% in cars and 20% in trucks [6]. Following another approach, the Eastern Research Group (2015) used EPA’s MOtor Vehicle Emission Simulator (MOVES) along with an empirical compliance factor obtained from historical data of the I/M program implemented in Austin and they estimated VOC, CO and NO_X_ emissions reductions of 11.3%, 12.7% and 6.3%, respectively [7].

Nonetheless, it is more challenging to estimate the emissions reduction that an I/M could achieve when it has not been implemented. A few studies have intended this objective but have been limited to complex laboratory tests on small samples of vehicles reproducing the I/M procedure [8] or to emissions models predictions based on suggested emission and deterioration factors that may not represent their local conditions [9]. This work is aiming to address this issue by providing a relatively simple methodology, based on a single RSD campaign, to estimate the emissions reduction of pollutants as a consequence of implementing an I/M program in a region where it has not been implemented.

## 2. Materials and Methods

Aiming to estimate the percentage of reduction of tailpipe emissions that could be obtained by implementing an I/M program, a methodology based on three major steps (Figure 1) is proposed:The first step involves carrying out an RSD campaign of vehicles that are circulating under similar acceleration and speed conditions to those followed in the I/M program.The second step determines the overall conformity factor in vehicle emissions by using model-year compliance fractions (based on local regulations) obtained from the measurement campaign along with the public registry of the vehicles in the area.The third step estimates the emissions reductions derived from the I/M program by comparing the total mass of the pollutant generated by the non-complying vehicles for a year with the mass that would be emitted under the effect of an I/M program.

It is important to mention that this methodology is independent of the region of study. However, as seen in the previously mentioned studies [5,6,7], the results obtained will differ from region to region. The difference between results depends not only on the I/M test procedure imposed but also on the amount of non-complying vehicles that would be subject to mechanical maintenance. Additionally, non-complying vehicles not only differ in number between region to region, but also in the levels of pollution they generate. Zhang et al. (1995) studied the difference in vehicle emissions obtained through RSD in 22 different locations around the world [10]. He concluded that the average emissions of the fleet are dominated by a small percentage of highly polluting vehicles. Possible explanations for this discrepancy reported in other studies have included differences in the emissions regulations for new vehicles, the mechanical condition of circulating vehicles, local fuel properties, altitude, and the RSD monitoring conditions [11,12,13]. The understanding of the differences in the results between region to region will not be addressed in our work.

This methodology was implemented in Monterrey’s Metropolitan Area (MMA), which is a large semi-arid urban region located in northeast Mexico (near the southern border of the United States of America) with an average altitude of 550 m.a.s.l. (Figure 2). It is a region composed of 13 municipalities from the state of Nuevo León with a combined surface of 6794 km^2^. It is the third-most populous urban area in the country with more than 4 million inhabitants and it is the second-largest industrial region responsible for 9% of the national Gross Domestic Product (GDP) per capita [14]. There are 2.2 million vehicles registered in MMA, which makes it the region with the highest vehicle motorization index in Mexico (∼0.5 vehicles per inhabitant).

The emissions of vehicles in MMA are of great concern as they were recently reported as the main sources of CO (96%) and NO_X_ (68%) in the region [15]. In a previous study in MMA, the local environmental agency reported that old vehicles (>10 years) were the highest emitters, being responsible for the 67, 72, and 56 percent of the total vehicle CO, HC, and NO emissions, respectively. Additionally, it was also reported that MMA presented a large number of imported used cars (25%) [16]. This latter category is of particular interest as these vehicles are mostly old vehicles with high mileage, which usually translates into vehicles technologically obsolete with high emissions [17]. Despite the above mentioned, this region does not have an I/M program implemented.

For this case study, we worked with *Opus Inspection SA de CV* to measure 33074 vehicles using an OPUS RSD4600. After considering only light passenger gasoline vehicles within the device thresholds, a total of 17550 vehicles were successfully measured. The RSD technique (Figure 3) measures vehicle emissions by monitoring the light intensity depletion at characteristic wavelengths of a narrow infrared (NDIR) and ultraviolet (UV) beam of light that is casted across the road where vehicles transit. The measured emissions are then matched with the speed-acceleration (obtained with additional sensors) and with the vehicle characteristics obtained from the vehicular register office via the license plate, which is captured with a video camera system. The RSD technique has been widely used for identifying highly polluting vehicles [18].

The measurement campaigns were performed in March 2018 in 4 different MMA locations with diverse socioeconomic characteristics (Figure 2). Roads with high vehicular-flow and with a slight positive grade were considered to ensure adequate vehicle speeds and accelerations (Table 1). Only results from vehicles driven under vehicle speeds between 1 and 35 mph (∼1.6 to 55 km/h), positive accelerations between 0 and 4 mph/s (∼1.8 m/s${}^{2}$) and Vehicle Specific Power (VSP, that is, the power demanded to the engine) values between 3 and 20 kW/Metric-Ton were considered. These test conditions are similar to those followed in I/M programs conducted on chassis dynamometers [19]. With this consideration, 6476 valid measurements were obtained, which is equivalent to ∼0.4% of all the light passenger gasoline vehicles registered in the city, and 37% of the vehicles sampled.

Subsequently, the obtained data was compared with the tailpipe emissions limits fixed in the local regulations for in-use vehicles in order to evaluate vehicle emissions compliance. The current Mexican regulations [20,21] include three alternatives to identify vehicles that must undergo maintenance to keep their emissions under acceptable values—(i) On-Board Diagnostics (OBD) to identify malfunctioning after-treatment components, (ii) Emission tests on a chassis dynamometer, and (iii) RSD tests. For these last two alternatives, the Mexican regulations fixed the threshold values by model-year as shown in Figure 4a–c. In this sense, three sets of tailpipe emission limits included in the local I/M program were considered—(i) Nationwide limits [20] for tests conducted on a chassis dynamometer following the dynamic driving cycle PAS5024 & PAS2540 [22], which is based on the Acceleration Simulation Mode (ASM) test procedure followed in I/M programs in the United States of America [23]; (ii) Stricter limits for large urban centers (megacities) like Mexico City and Guadalajara following the same test procedure [21]; and (iii) Limits to identify high emitters using RSD tests [21]. For each case, the obtained fraction of vehicles that do not comply with the regulation was reported.

Assuming that the sample of vehicles measured is representative of the MMA vehicular fleet, the percentage of registered vehicles that would not comply with the emissions limits (*I*) is estimated using Equation (Equation 1). In this equation *N* represents the total number of registered vehicles in the region, *n_i,j_* is the number of registered vehicles of model *i* (e.g., Nissan Sentra) and model-year *j*, and *q_i,j_* is the non-compliance percentage obtained from the RSD measurement campaign for vehicles of model *i* and model-year *j*.
(1)$$I=\frac{1}{N}\sum_{i}\sum_{j}q_{i,j}n_{i,j}.$$

Thereafter, the data from the RSD campaign was used to obtain fuel-based emission factors (mass of pollutant emitted per unit of fuel) and to estimate the potential reduction in emissions that could result from implementing the Mexican I/M program in MMA. It is important to clarify that the obtained emission factors are used for comparison purposes only and not for emissions inventory, as the measurement conditions were intended to simulate I/M conditions and not to represent the local driving patterns.

## 3. Results and Discussions

### 3.1. Vehicles Emissions Compliance

Table 2 shows the overall fleet concentrations obtained after data quality analysis and measurement conditions filtering. Figure 4a–c show the Mexican limits for in-use vehicles along with the pollutant concentration results obtained in the measurement campaign, organized by model-year of the vehicles for CO, HC (expressed as hexane equivalent), and NO. In these figures, the boxes indicate the first and third quartiles. The open circles are the outliers (values over 1.5 times the interquartile range), and the *x* and *–* symbols specify the mean and median values, respectively. The results show that the mean concentrations increase with the vehicles’ age. This result is well known in the literature as an effect of the evolution of model-year vehicle technologies [24]. However, it was also noted that there is an increment in the data dispersion (boxes sizes and whiskers) along with a rising gap between the model-year mean and median concentrations with the vehicles’ age. Both are caused by the continuous increment of highly polluting vehicles (outliers), which is expected for urban centers where no in-use emission control programs are applied.

These results are now used to estimate the fraction of vehicles that do not comply with the local regulation for in-use vehicles (Table 3), which would be subject to a mechanical maintenance program. When comparing the RSD results obtained in the monitoring campaign to the limits established in the local RSD regulation, it was concluded that 17.7% of the light-duty vehicles circulating in MMA are allegedly high-polluting vehicles, that is, 17.7% of the vehicles measured in the RSD campaign surpass at least one of the CO, NO or HC threshold values specified in the local regulation. This fraction changes to 17.2% when only the locally registered vehicles are considered (Equation (Equation 1)). The difference in these percentages is due to the influence of the large fraction of visiting vehicles in MMA. Nevertheless, the preferred method in Mexico to identify vehicles that must undergo maintenance is the tailpipe emissions measurements on a chassis dynamometer. The local regulation establishes stricter threshold values for this method compared to those established for the RSD method. Assuming that both methods produce equivalent results, it was estimated, using Equation (Equation 1), that 39.0% of the vehicles registered in MMA surpass at least one of the threshold values specified in the local regulation for CO, NO, and HC for tests on a chassis dynamometer and consequently must undergo maintenance. Table 3 details the percentage of vehicles that surpass each of the emissions’ thresholds independently.

### 3.2. Equivalence between RSD and the I/M Program

The previous results depend on the validity of the assumption that RSD and I/M tests produce equivalent results. Stedman et al. (1997) confirmed it to a certain extent by evaluating the equivalency between RSD and the IM240 test [25]. Aiming to validate this hypothesis for the particular conditions in Mexico, their approach was replicated. To this end, 105071 measurements taken by the Mexico City environmental authority during July 2018 from their PAS5024 & PAS2540 I/M program were used and compared to the 6476 RSD measurements obtained from our 2018 MMA campaign.

This comparison also included 8110 RSD measurements obtained through another RSD campaign that was carried out in Mexico City so that the potential effects on vehicle emissions of the particular conditions of these cities could be taken into account. The Mexico City RSD campaign was carried out by the National Institute of Ecology and Climate Change (INECC) following the same procedure stated in Section 2, between January and June 2017. A total of 17 different locations were measured between the hours of 10:00 and 17:00. Taxicabs were excluded from this comparison as it is a common practice for these vehicles to perform their mechanical maintenance just before their periodical I/M program, which would lead to higher discrepancies in emissions between both RSD and I/M test procedures.

Figure 5a,b show the linear correlation obtained through the direct comparison between the RSD and the PAS5024 & PAS2540 results for different vehicle technologies and model-years. It was discovered that both methods produce results that are highly correlated for all pollutants in both cities (R^2^ > 0.87). There is a lower discrepancy in emissions when considering the measurements from Mexico City only. Yet their RSD concentrations are still around 5, 3, and 9 times higher than those obtained on the chassis dynamometer tests for CO, NO, and HC, respectively.

Additionally, to reduce the effect of the different model-year vehicle technologies and mechanical conditions on the emissions discrepancy between the RSD and I/M program, a single technology (same vehicle model) with less than a year of use was considered in the comparison shown in Figure 6. The vehicle used for this comparison is a medium passenger vehicle (NHTSA classification) with a 1.6-liter atmospheric engine. This vehicle model has the most common vehicle configuration in Mexico according to data obtained from the Secretariat of Communications and Transportation (SCT). A total of 396 vehicles of the same model and model-year were included in this comparison.

The results do not exhibit a normal distribution. Therefore, non-parametric tests are required for comparing differences in pollutant concentrations. The Mood’s Median Test was carried out for this purpose. Results show that there is not enough evidence to conclude that the differences between the corresponding CO, HC, and NO medians of this single vehicle model are statistically significant between the Mexico City I/M test and the Mexico City RSD campaign. The MMA concentrations present discrepancies with respect to the Mexico City PAS5024 & PAS2540 concentrations, which could be related to differences in fuel quality, altitude, and in-use emissions control strategies. The results may not be equivalent in every case between RSD and PAS5024 & PAS2540 technique as the measurement conditions differ. Yet, our analysis for a widely common vehicle model in this region has proven that the results are similar.

### 3.3. Potential Reduction in Vehicles’ Emissions

Aiming to estimate the emission reduction of implementing an I/M program, the total mass of the pollutants generated by the non-complying vehicles should be compared against the mass that would be emitted by those vehicles under the effect of an I/M program. The total mass of pollutant *k* generated by the vehicles in a year (*E_k_*) is calculated using Equation (Equation 2), where *E_f_i,j,k__* is the emission factor of the pollutant *k* for vehicles of model *i* and model-year *j* (in g/ *VKT*) and *VKT_i,j_* are the average kilometers traveled in one year by vehicles of model *i* and model-year *j*.
(2)$$E_{k}=\sum n_{i,j}E_{f_{i,j,k}}VKT_{i,j}$$

Even though the existence of I/M programs promote the renovation of the vehicle fleet (varying *n_i,j_*), and therefore the *VKT_i,j_*, the effects of these variations on *E_k_* are negligible when compared to the variations on the emission factors (*E_f_i,j,k__*). Thus, the study focused on the effects of the I/M program on the emission factors. The emission factors are obtained experimentally by measuring the mass of pollutants emitted from a large sample of vehicles operating under normal conditions for a long time. Extensive work has been conducted with this objective in mind. The USEPA has collected the most representative results from several studies in the United States of America and included them in MOVES. Similarly, the European community has done it with COPERT. Both report their emission factors per bins of speed-acceleration. It is unknown if those emission factors can be used for the case of MMA. Therefore, it was proposed as an approximation to assume that the real emission factors of the vehicles in MMA (*E_f_i,j,k__*) are proportional to the emission factors measured during the RSD campaign (*I_k_*). This approach cannot be used to estimate the total mass pollutants emitted; however, it can be used to estimate the percentage of reduction in the mass of pollutants emitted as an effect of an I/M program, without knowing the true emission factors.

RSD data can be used to obtain fuel-based emission factors (*I_k_**) [26]. These factors are calculated through a carbon balance in the fuel and combustion products, as seen in Equation (Equation 3). In this equation *I_k_** is the mass of pollutant *k* emitted per unit of fuel, *Q_k_* is the measured volumetric fraction between pollutant *j* and CO_2_, *M_k_* is the molecular weight of pollutant *k*, ρ_*f*_ is the fuel density, and *X_c_* is the carbon mass fraction contained in the fuel. A fuel density of 700 kg/m${}^{3}$ and an *X_c_* of 0.85 were used, which are the values reported by the fuel manufacturer and distributor in MMA (PEMEX). By multiplying *I_k_** by the specific fuel consumption of the vehicle, the mass of the pollutant *k* emitted per kilometer traveled (*I_k_*) can be obtained. The obtained distributions of *I_k_** for CO, HC and NO are presented in Figure 7a–c, respectively. The results are expressed as the mean value per model-year with their respective standard error (vertical bar). Table 3 reports the average values for *I_k_** obtained in our study. It is important to emphasize that results were obtained under the RSD monitoring campaign conditions mentioned in Section 2 and not under the normal vehicle operation in this region. Therefore, the *I_k_** are not the actual emission factors representative of the emission of the vehicles in MMA.
(3)$$I_{k}^{*}=\frac{Q_{k}}{1+Q_{CO}+6Q_{HC}}\frac{M_{k}}{12}\rho_{f}X_{c}.$$

Finally, using these results and Equation (Equation 2), the reduction (expressed as a percentage) in the mass emission of pollutants that would be obtained through the implementation of an I/M program in MMA was estimated (Table 4). As a conservative scenario, it was assumed that after a maintenance service, the emissions of the vehicles that exceeded the emission standards will be taken to the maximum permissible limit for each pollutant. Table 4 shows that, under these assumptions, the implementation of an I/M program in MMA will potentially reduce the current vehicular emissions of CO, HC, and NO in approximately 42%, 69%, and 28%, respectively.

### 3.4. Real Fuel Consumption in MMA

When the mass of pollutant *k* needs to be estimated through Equation (Equation 2), the fuel-based emission factors (*I_k_**) must be divided by the fuel economy (km/L) or multiplied by specific fuel consumption (*SFC* in L/km) to convert it into the emission factor (*E_f_i,j,k__*).

Real values of vehicles’ fuel economy were obtained from the instrument panel of 372 vehicles (model-years between 2008 and 2018) in a new campaign carried out in gas stations within MMA. Figure 8 compares the values obtained with those reported by the vehicle manufacturers for brand new vehicles following type-approval tests. It was found that in-use light and medium-duty vehicles reported similar values of fuel economy between each-other, exhibiting a normal distribution.

It was also observed that the obtained fuel economy is on an average 33 percent lower to the values reported by the vehicle manufacturers. Furthermore, the real fuel economy presented a low correlation with the manufacture year (R^2^ < 3.6 percent), accumulated traveled distance (R^2^ < 2.66 percent) and engine displacement of the vehicle (R^2^ < 25.12 percent), indicating that it depends more on the driving pattern of the area than on these factors. Then, it was concluded that the best estimate of the real fuel economy and kilometers traveled per year in MMA are the average values obtained in this additional campaign, which resulted in 11.42 ± 4.96 km/L and 14,240 km/year, respectively.

### 3.5. Reducing Vehicles Emissions

I/M programs have proven to be an effective strategy to control the excessive vehicle emissions related to unattended mechanical conditions. In Section 3.3 we obtained, under a conservative scenario, the I/M potential effect on reducing the MMA on-road vehicle emissions. However, its impact on improving the air quality in cities depends on the share of vehicles powered by Internal Combustion Engines (ICE) with fossil fuels.

It’s important to clarify that I/M programs help controlling ICE vehicles emissions exclusively, as they exclude electric vehicles due to the nature of the program of evaluating tailpipe emissions only. Recent smart mobility strategies such as ridesharing [27] within the concept of Mobility as a Service (MaaS) and the improvement of the city infrastructure for electric vehicles [28] have proven to be very effective approaches to improve the transportation network, hence reducing the use of ICE vehicles and increasing the electric vehicle market worldwide. Consequently, when evaluating strategies to reduce the vehicle emissions in cities with a substantial number of electric vehicles, the environmental cost of electric vehicles should also be considered as it is a notable problem in today’s emissions related to ground transportation [29].

## 4. Conclusions

Inspection and Maintenance (I/M) programs were created with the objective of improving air quality by periodically detecting highly polluting vehicles due to inadequate mechanical maintenance. In this work, with the objective of helping local authorities of non-I/M areas to decide if I/M programs are necessary and adequate for their particular conditions, a methodology, based on local on-road data, to estimate the potential emissions reductions of an I/M program is proposed. As a case study, this methodology was followed in Monterrey’s Metropolitan Area (MMA), which is a non-I/M area with more than 4 million inhabitants and 2.2 million vehicles. To obtain the local on-road data, an RSD campaign was carried out in March 2018 in four different sampling locations within MMA. Approximately 0.4% of the registered vehicles in the region were sampled under similar conditions to those found in the I/M test. The measured emission concentrations were compared against the emission limits established in the national emission regulation for in-use vehicles. It was found that 39.0% of the vehicles registered in MMA would not comply with this regulation and that 17.2% are allegedly highly polluting vehicles.

Fuel-based emission factors were obtained from the RSD campaign along with an average fuel economy obtained with an additional campaign on 372 vehicles. With this additional campaign, it was concluded that in MMA, light and medium-duty gasoline-fueled vehicles exhibit a fuel economy of 11.42 ± 4.96 km/L and are used on average 14240 km/year. The real fuel economy is, on average, 33 percent lower than the fuel economy reported by the manufacturers for brand new vehicles from type-approval tests.

These values allowed an estimation of the emission mass generated by the vehicles in a year. These results were compared to those potentially obtained when non-compliance vehicles are forced to undergo an I/M program. This work led to the conclusion that under a conservative scenario, the implementation of an I/M program in MMA would potentially reduce the current vehicle emissions of CO, HC, and NO in approximately 42%, 69% and 28%, respectively.

## Figures and Tables

**Figure 1 ijerph-17-04730-f001:**
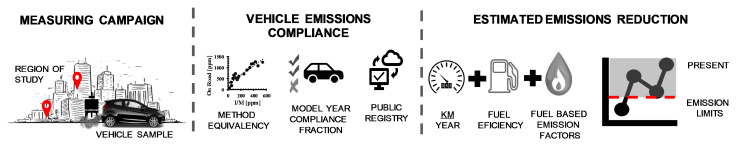
Illustration of the proposed methodology to estimate the percentage of reduction of tailpipe emissions that could be obtained by implementing an I/M program.

**Figure 2 ijerph-17-04730-f002:**
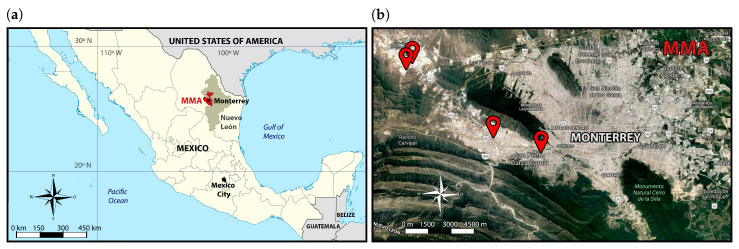
Detailed map on the location of (**a**) Monterrey’s Metropolitan Area (MMA) and (**b**) the respective measurement sites.

**Figure 3 ijerph-17-04730-f003:**
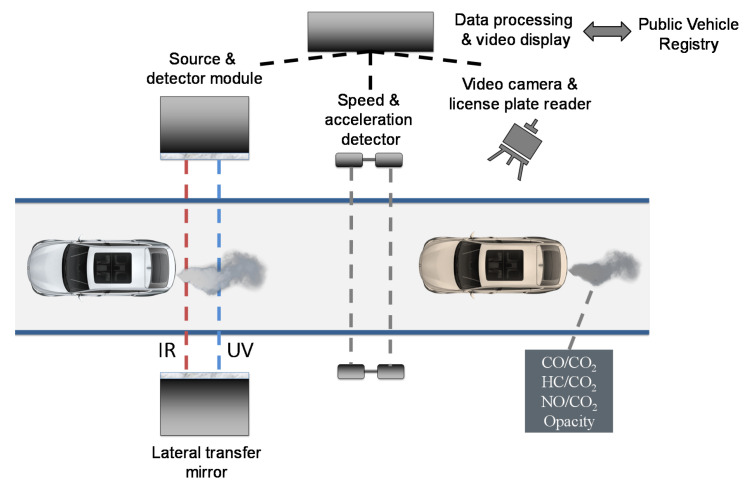
Remote Sensing Device (RSD) technique illustration. *Source:* Authors’, based on Reference [18].

**Figure 4 ijerph-17-04730-f004:**
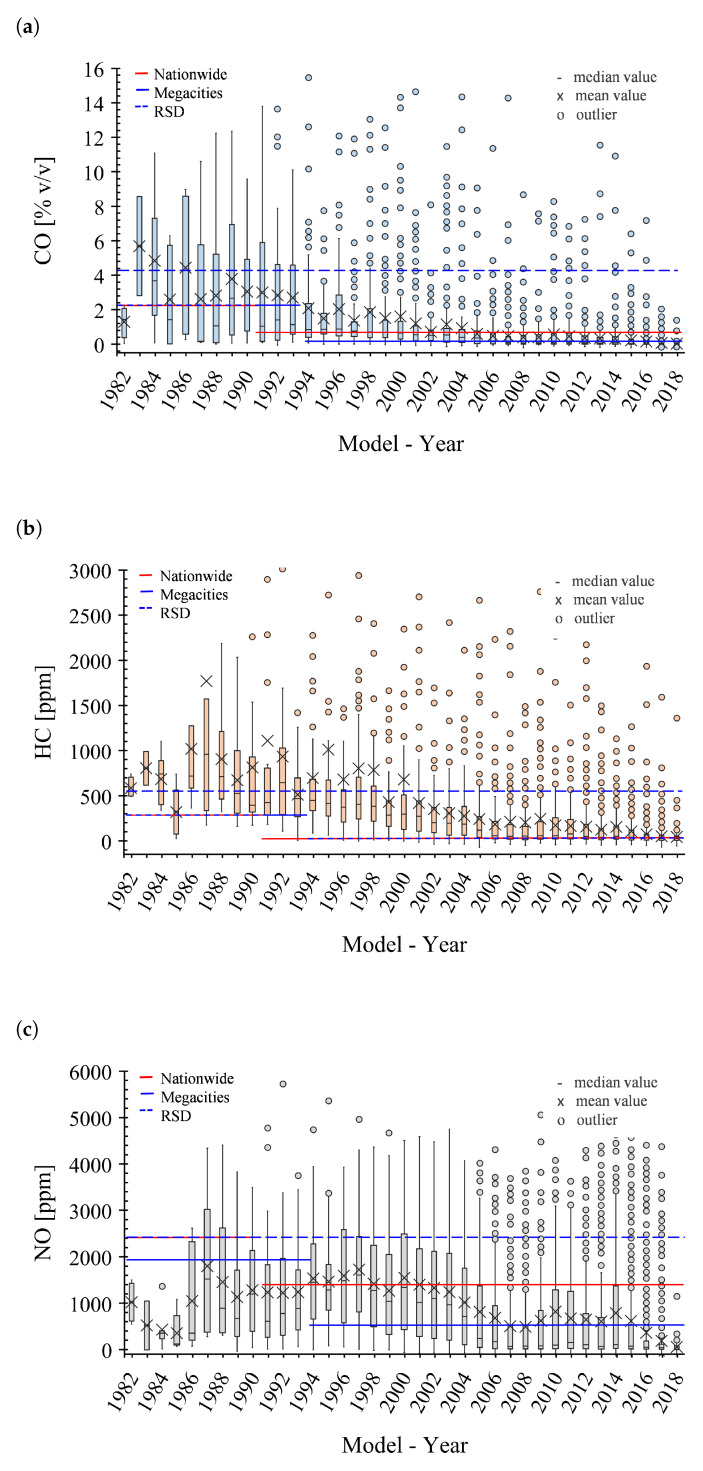
Boxplots of the (**a**) CO, (**b**) HC (hexane equivalent), and (**c**) NO tailpipe concentrations, per model-year, measured during the 2018 MMA RDS monitoring campaign.

**Figure 5 ijerph-17-04730-f005:**
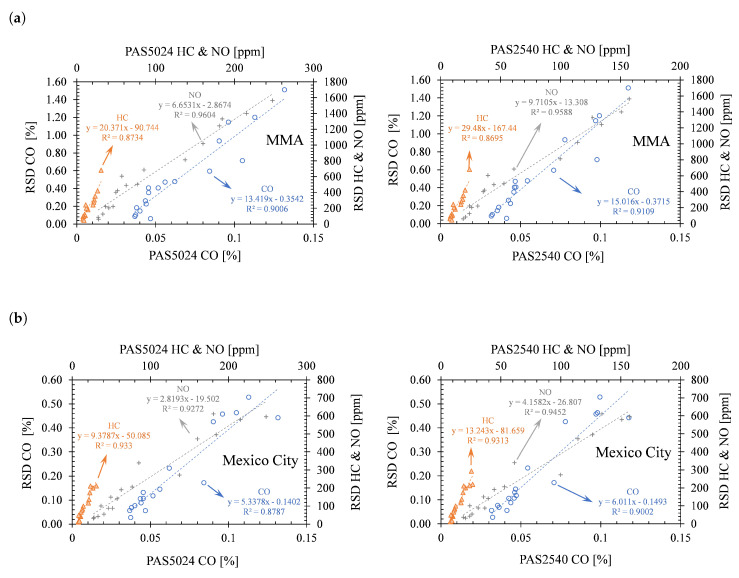
Correlation analysis between the CO, HC (hexane equivalent), and NO tailpipe mean model-year concentrations from RSD and the ASM I/M program (PAS5024 & PAS2540) for (**a**) MMA and (**b**) Mexico City. (Period 2000–2017).

**Figure 6 ijerph-17-04730-f006:**
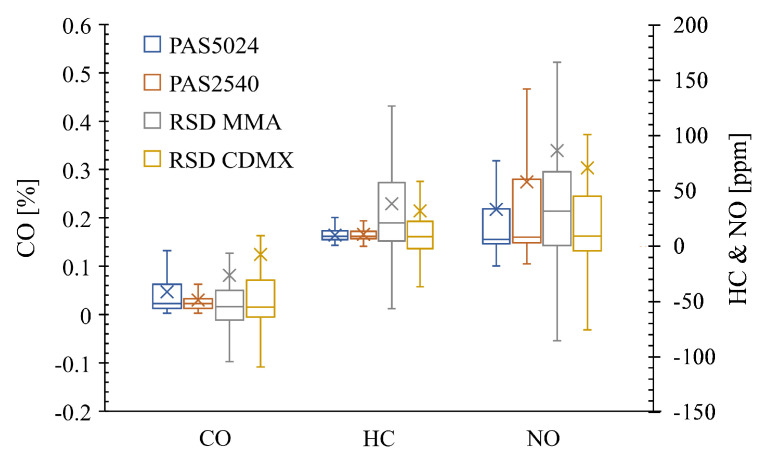
Boxplots of the CO, HC (hexane equivalent), and NO tailpipe concentrations for the one-year-old single model from RSD and the ASM I/M program (PAS5024 & PAS2540) for MMA and Mexico City.

**Figure 7 ijerph-17-04730-f007:**
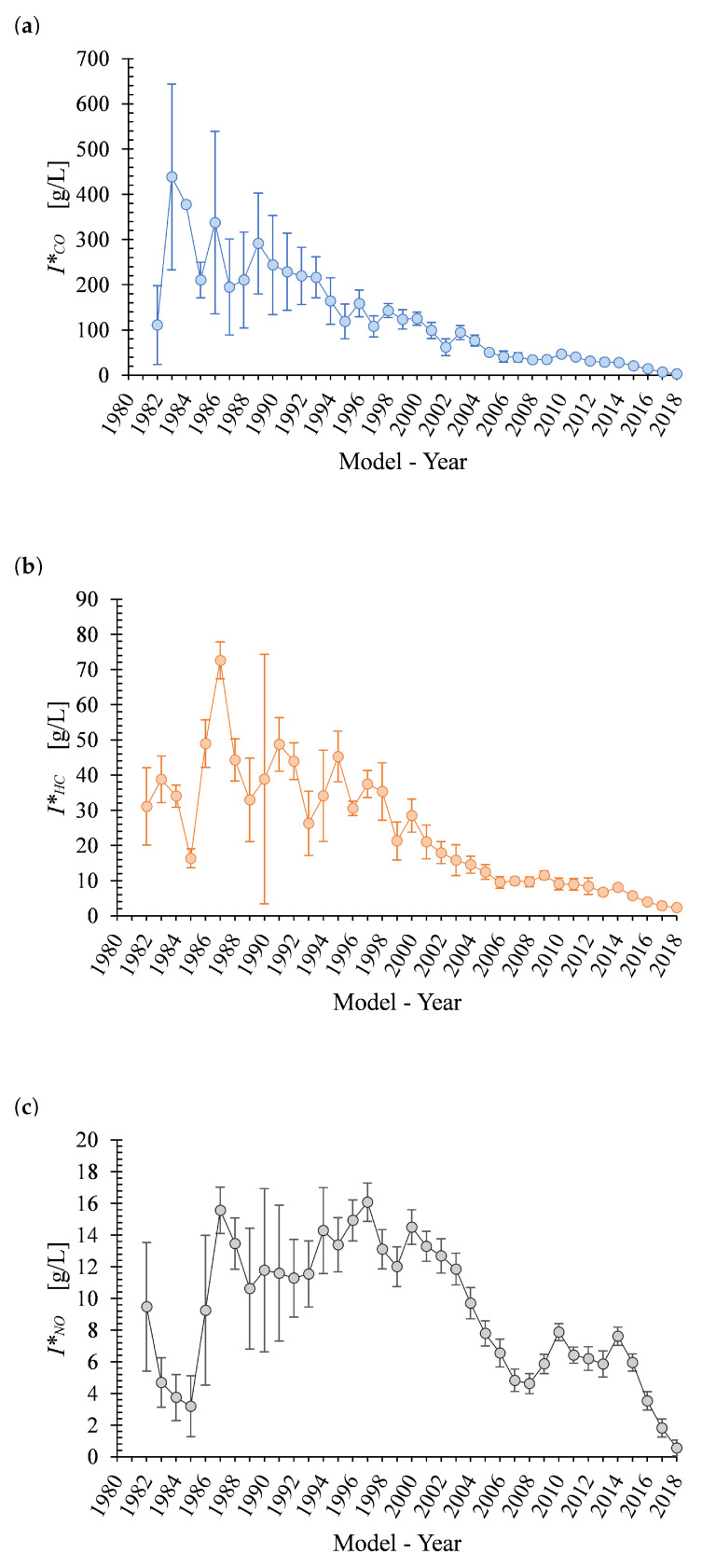
MMA fuel-based emission factors (${I_{k}}^{*}$) for (**a**) CO, (**b**) HC (propane equivalent), and (**c**) NO.

**Figure 8 ijerph-17-04730-f008:**
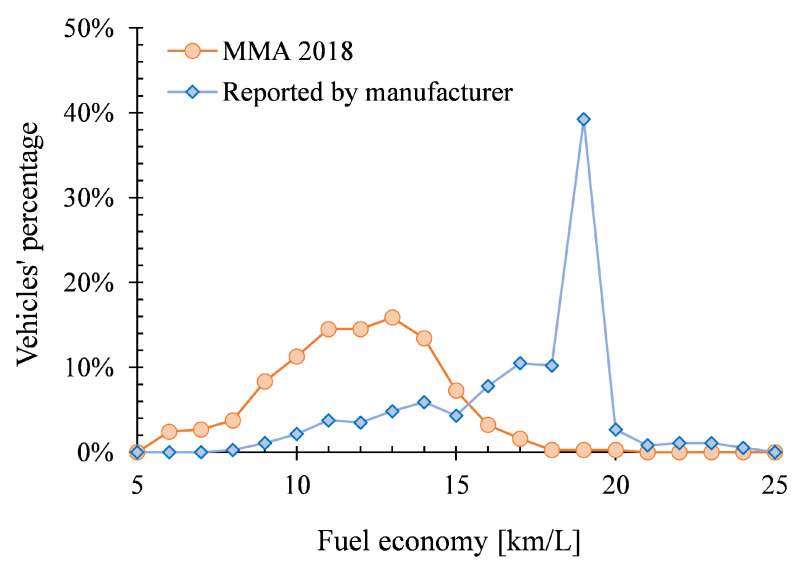
Comparison between the fuel economy reported by manufacturers and the real values obtained from circulating vehicles in MMA.

**Table 1 ijerph-17-04730-t001:** Monitoring campaign carried out in MMA in 2018.

Municipality	Address	Category	Date	Road Grade [Degrees]
San Pedro	Manuel Santos 540	R	5 March 2018	0.4
Santa Catarina	Callejón del Mármol 215	R/C/I	6 March 2018	0.3
García	Sierra Real 66000	I	7 March 2018	1.7
García	Heriberto Castillo 1000	I	7 March 2018	3.2

R = residential; C = commercial; I = industrial.

**Table 2 ijerph-17-04730-t002:** Overall results of the MMA RSD campaign.

	MMA
Date	March 2018
Campaign schedule	07:00 to 19:00
Total Measurement attempts	33,074
Vehicles Measured	17,550
Valid measurements for CO, HC & NO	13,124
Licensed plate matched	10,434
Adjusted to speed and Acceleration ranges	7485
Adjusted to VSP range	6476
Avg./Median Fleet age [years]	8.8/6
Avg./Median Speed [mph]	16.8/16.3
Avg./Median Acceleration [mph/s]	0.93/0.83
Avg./Median VSP [kW/Ton]	6.76/6.15
Avg. Slope [Degrees]	1.4
Mean ^a^/Median CO [vol %]	0.598 ± 0.018/0.093
Mean ^a^/Median HC ^b^ [ppm]	232 ± 8/62.5
Mean ^a^/Median NO [ppm]	705 ± 13/123.2
Mean ^a^/Median CO [g/L]	49.1 ± 0.8/8.5
Mean ^a^/Median HC ^c^ [g/L]	11.5 ± 0.2/3.5
Mean ^a^/Median NO [g/L]	6.7 ± 0.1/1.2

^a^ Expressed with standard error; ^b^ hexane; ^c^ propane.

**Table 3 ijerph-17-04730-t003:** Emissions’ thresholds from local regulations for in-use vehicles along with the estimated percentage of non-complying vehicles in MMA, according to the 2018 RDS monitoring campaign

	Emissions’ Thresholds for in Use Vehicles					Non-Complying Vehicles			*q_i,j_* Non-Complying Vehicles		
	CO [%v/v], NO & HC [ppm]					(Measured) [%]			(Registered) [%]		
	**On Dynamometer**				**RSD**	**On Dynamometer**		**RSD**	**On Dynamometer**		**RSD**
	**National ${}^{1}$**		**Megacities ${}^{2}$**			**National ${}^{1}$**	**Megacities ${}^{2}$**		**National ${}^{1}$**	**Megacities ${}^{2}$**	
	≤**1990**	≥**1991**	≤**1993**	≥**1994**							
CO	2.5	1.0	2.5	0.7	4.5	12.6	19.6	3.2	14.3	19.3	4.4
NO	2500	1500	2000	700	2500	19.6	29.7	9.5	16	23.9	6.8
HC	350	100	350	100	600	40.6	40	7.9	37.5	36.7	10
Global	N/A	N/A	N/A	N/A	N/A	43.5	47	17.7	39	40.9	17.2

${}^{1}$ National regulation for in-use vehicles by model-year following the PAS5024 & PAS2540 test [20]. ${}^{2}$ Regulation for in-use vehicles by model-year in large urban regions following the PAS5024 & PAS2540 test [21]. RSD: Regulation for in-use vehicles following a Remote Sensing Device test [21]. N/A: Not applicable.

**Table 4 ijerph-17-04730-t004:** Potential percentage of reduction of tailpipe HC, CO and NO emissions from vehicles in MMA as a result of the implementation of an I/M program.

Test Protocol Considered	Regulation Considered	Potential Reduction [%]		
		CO	HC	NO
On chassis dynamometer	National ${}^{1}$	42	69	28
	For large urban centers ${}^{2}$	47	67	55
RSD		10	25	9

${}^{1}$ NOM-041 [20]; ${}^{2}$ NOM-167 [21].

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
