# Peer review of "Assessment of the Reduction in Vehicles Emissions by Implementing Inspection and Maintenance Programs"

_ijerph, 2020, doi:10.3390/ijerph17134730_

Round 1

Reviewer 1 Report

The topic is worthy of investigation, and the paper is quite well-written and structured. However, I suggest to carefully revise the English language and the punctuation.

Table 2 is not very clear. I suggest to revise it.

Figure 2 could be divided into two sub-figures: figure 2a and figure 2b, where in figure 2a Mexico is showed, and in figure 2b Monterrey’s Metropolitan Area.

Reviewer 2 Report

The authors provided a novel and interesting methodology to measure the potential environmental impact of I/M programs. The concerned issues are listed as follows: 1. Lines 24-29. The differences between the results of the literature are quite significant. You’re your point of view, what has caused these differences? Will this problem be addressed by your methodology? 2. The electric vehicles (Lin and Greene, 2011) and the research in their environmental costs (Ma et al., 2020a) are quite popular in recent years. I noticed that your methodology only examined the emissions directly produced by the vehicle itself. How to consider the environmental costs of electric vehicles? Lin, Z., Greene, D.L., 2011. Promoting the market for plug-in hybrid and battery electric vehicles: role of recharge availability. Transportation Research Record: Journal of the Transportation Research Board 2252, 49–56. https://doi.org/http://dx.doi.org/10.3141/2252-07 Ma, J., Wang, H., Tang, T., 2020a. Stochastic Electric Vehicle Network with Elastic Demand and Environmental Costs. Journal of Advanced Transportation 2020, 1–11. https://doi.org/10.1155/2020/4169826 3. Lines 72-73. Please make some analysis of what has caused the low sample rate (17550/33074). Does it suggest a limited application scope of this methodology? If so, how to improve it. 4. I noticed that this methodology only takes into account the number of vehicles rather than the types of vehicles or the amount of travel demand. Specifically, a bus, a ridesharing vehicle, or a carpooling vehicle may emit a little more because it takes a heavier weight. However, it takes more travelers and meets more travel demand (Ma et al., 2020b). Moreover, ridesharing or carpooling vehicles even belong to the same type as a conventional vehicle. How to measure these vehicles so that we can encourage the travel modes with higher vehicle occupancy rates. Ma, J., Xu, M., Meng, Q., Cheng, L., 2020b. Ridesharing user equilibrium problem under OD-based surge pricing strategy. Transportation Research Part B: Methodological 134, 1–24. https://doi.org/10.1016/j.trb.2020.02.001 5. Eq. (1). As time goes on, will it cause bias to use the model-year rather than a current year? Or we can say, does it means that the compliance factor needs to change year by year?

Round 2

Reviewer 2 Report

I can see that the authors have made their best to modify this manuscript. Their efforts are much of appreciate. However, some of their defense is too thin. Details are given as follows.

  1. About Comment 2 of the last version. We know that there are not many electric vehicles in MMA, and perhaps all around the world. However, we can see that in a near future, the use of electric vehicles may be popular since many government around the world have made their best to promote electric vehicles. Since scientific research needs to be forward-looking, I strongly suggest that the authors cite the mentioned research papers on the environmental cost of electric vehicles in this manuscript and consider the environmental impact of electric vehicles as a future work.
  2. About Comment 4 of the last version. Although it is hard to consider ridesharing and travel demand in this manuscript, it is still a proven fact that ridesharing is gaining more and more attention and changing people’s travel behavior. Moreover, the final purpose of studying environmental impact is to improve the environment, right? Designing a methodology to encourage the use of ridesharing has a potential to achieve it. Since the authors also consider it is a worthy focus for a further study but is not the current focus of this present work, I strongly suggest the authors to cite some works on the ridesharing problems and highlight the transportation network impact of ridesharing.   
  3. Other concerned issues have been well addressed.
